# Overestimation of the effect of climatic warming on spring phenology due to misrepresentation of chilling

Huanjiong Wang [1,2 ✉], Chaoyang Wu [1,2 ✉], Philippe Ciais [3], Josep Peñuelas [4,5], Junhu Dai[1,2], Yongshuo Fu[6] & Quansheng Ge [1,2 ✉]

Spring warming substantially advances leaf unfolding and flowering time for perennials. Winter warming, however, decreases chilling accumulation (CA), which increases the heat requirement (HR) and acts to delay spring phenology. Whether or not this negative CA-HR relationship is correctly interpreted in ecosystem models remains unknown. Using leaf unfolding and flowering data for 30 perennials in Europe, here we show that more than half (7 of 12) of current chilling models are invalid since they show a positive CA-HR relationship. The possible reason is that they overlook the effect of freezing temperature on dormancy release. Overestimation of the advance in spring phenology by the end of this century by these invalid chilling models could be as large as 7.6 and 20.0 days under RCPs 4.5 and 8.5, respectively. Our results highlight the need for a better representation of chilling for the correct understanding of spring phenological responses to future climate change.

[1] Key Laboratory of Land Surface Pattern and Simulation, Institute of Geographic Sciences and Natural Resources Research, Chinese Academy of Sciences, Beijing 100101, China. [2] University of Chinese Academy of Sciences, Beijing 100049, China. [3] Laboratoire des Sciences du Climat et de l'Environnement, IPSL-LSCE CEA CNRS UVSQ, 91191 Gif sur Yvette, France. [4] CSIC, Global Ecology Unit CREAF-CSIC-UAB, Bellaterra, Barcelona 08193 Catalonia, Spain. [5] CREAF, Cerdanyola del Valles, Barcelona 08193 Catalonia, Spain. [6] College of Water Sciences, Beijing Normal University, Beijing 100875, China. ✉email: wanghj@igsnrr.ac.cn; wucy@igsnrr.ac.cn; geqs@igsnrr.ac.cn

Spring phenology (e.g., budburst, leaf-out, and flowering) of perennials has advanced in recent decades around the middle and high latitudes (north of 30° N) of the Northern Hemisphere[1–3]. Shifts in spring phenology have essential implications for ecosystems and the climatic system, ranging from interactions among plants and their herbivores[4,5] to the balance among surface carbon, energy, and water[6–8]. Projecting future phenological changes are thus increasingly important for assessing the impact of climate change on ecosystems. Phenological prediction, however, still has large uncertainties, because climatic warming has dual effects on spring phenology[9]. Spring warming (forcing temperature) advances spring phenological events[10], but this effect is counteracted by the reduced chilling due to winter warming[11,12]. Several forcing and chilling models have been developed based on controlled experiments or inverse modeling (calibrated the model with the observation data) to quantitatively describe the effects of temperature on spring phenology[9]. A widely used forcing model based on growing degree days (GDDs)[13] calculates the accumulated number of degrees above a particular temperature threshold after a specific date. The generality of chilling models, however, limits operational applicability, because they were developed for specific species and climatic zones[14].

The dual role of temperature in regulating the spring phenology of perennials can be described by a negative relationship between the heat requirement (HR, the accumulated forcing temperature required for a phenological event) and the chilling accumulation (CA, the amount of chilling received by plants during endodormancy) (see the references in Supplementary Table 1). Experimental evidence suggests that saplings and twigs need less time and fewer degree days to budburst if pre-burst chilling in natural or controlled environments is longer[15,16]. Phenological observations also support this phenomenon when comparing HR with CA (based on several chilling models) in time and space[12,17,18]. From a detailed literature survey, we found that among 95 perennials investigated, 91 (95.8%) belonging to 46 genera followed this relationship (Supplementary Fig. 1, Supplementary Table 2). These species involve many life forms (herbs, deciduous or evergreen coniferous trees, and deciduous broad-leaved trees or shrubs), and include the most dominant forest tree species in Europe (Supplementary Fig. 2). The negative correlation between CA and HR is therefore common among perennials and stable across studies, species, and locations (Fig. 1). The molecular basis of the control of budburst mediated by chilling was recently identified for aspen[19]. Chilling temperatures down-regulate the SVL and TCP18 genes, which are negative regulators of budburst, and promote budburst under subsequent forcing conditions[19].

Many chilling models have been used to measure the amount of chilling by quantifying the rate of chilling in daily (or hourly) temperatures[20,21], but whether or not current chilling models can describe this physiological process is unclear. We use long-term in situ observations of leaf unfolding or flowering for 30 perennials at 15,533 phenological stations across central Europe (see the distribution of the stations in Supplementary Fig. 3 and the list of species in Supplementary Table 3) to compare long-term trends in CA and the relationship between CA and HR based on different chilling models. We aim to assess the ability of current chilling models to accurately simulate physiological processes and estimate uncertainties for predicting future changes in spring phenology. We show that 7 of 12 current chilling models fail to account for the correct relationship between CA and HR, leading to substantial overestimates of the advance of spring phenology under climate change. There is an urgent need to address the representation of phenological modeling, in this case specifically chilling models.

## Results

**Linear trends in CA**. We assessed 12 chilling models coded as $C_1$–$C_{12}$ (see "Methods" for a full description). The linear trends in CA during winter to early spring (previous November to April) from 1951 to 2018 were calculated for each station based on all models and E-OBS gridded temperature data set. We expected that the accumulated chilling in Central Europe has progressively decreased as the winter temperature (Previous November to February) increased significantly by 0.25 °C/decade from 1951 to 2018. The trends in CA and its spatial pattern, however, varied among the models (Fig. 2). CA for Models $C_1$, $C_2$, $C_4$, $C_5$, and $C_{12}$ decreased significantly at >88% of the stations ($p < 0.05$) because the winter temperature exceeded the maximum effective temperature under climate warming (e.g., 5 °C for model $C_1$). The proportion of stations reporting a significant decrease in CA for the other models only ranged from 1.0% (Model $C_8$) to 61.5% (Model $C_3$). The increasing trends even dominated across Central Europe for models $C_7$–$C_9$, possibly because the freezing temperature became effective under climate warming in these models, and this effect of the increase was higher than the effect of decrease causing by the loss of maximum effective temperature. Pearson's $r$ between each pair of chilling models for each station (Supplementary Fig. 4) indicated that not all chilling models were positively correlated with each other. Some models were even negatively interrelated (e.g., Model $C_1$ and $C_8$), suggesting that the current chilling models could not reflect a consistent trend when measuring the change in the amount of chilling.

**Relationship between CA and HR**. We calculated CA (from 1 November in the previous year to the date of onset of spring events) for each species, station, and year using the 12 chilling models to determine if the models met the physiological assumption of Fig. 1. Also, HR was calculated based on a commonly used forcing model (integrating daily mean temperatures >0 °C from 1 January to the date of onset of spring events). The correlations between CA and HR varied among the chilling models for all species. Figure 3 shows an example of leaf-out for *Betula pendula*. Five chilling models ($C_1$, $C_2$, $C_4$, $C_5$, and $C_{12}$) that predicted decreasing trends in CA exhibited significant negative correlations between CA and HR, and the other models exhibited positive correlations. Twenty-nine of the 30 species had significantly negative Pearson's $r$ between HR and CA based on the above five models (Supplementary Fig. 5). We therefore classified models $C_1$, $C_2$, $C_4$, $C_5$, and $C_{12}$ as valid models, because they met the physiological assumption in Fig. 1. No or only a few species had negative CA–HR relationships in the other seven models (Supplementary Fig. 5), which were thus classified as invalid models. We also analyzed the station-level relationship between CA and HR for all stations with at least 15-years of records. Few stations (<10% for all species) had significantly negative Pearson's $r$ between CA and HR when using the invalid models, but >30% of the stations (for most of the species) had significantly negative $r$ between CA and HR when using the valid models (Supplementary Fig. 6). Since we only used one forcing model (Model $F_1$, see the equations in "Methods") to calculate HR, we further validated the chilling models by correlating them with HR based on the other seven forcing models (Models $F_2$–$F_8$). The percentage of stations with significantly negative Pearson's $r$ between CA and HR was larger for the valid than the invalid models, regardless of the forcing model chosen (Supplementary Fig. 7). Thus, the performance of the chilling models was not affected by the choice of forcing models. Although some of the invalid models were calibrated specifically for fruit and nut trees (e.g., Model $C_7$ and $C_9$), they still did not perform better than the valid models for *Prunus avium* (a fruit tree supplying edible cherry)

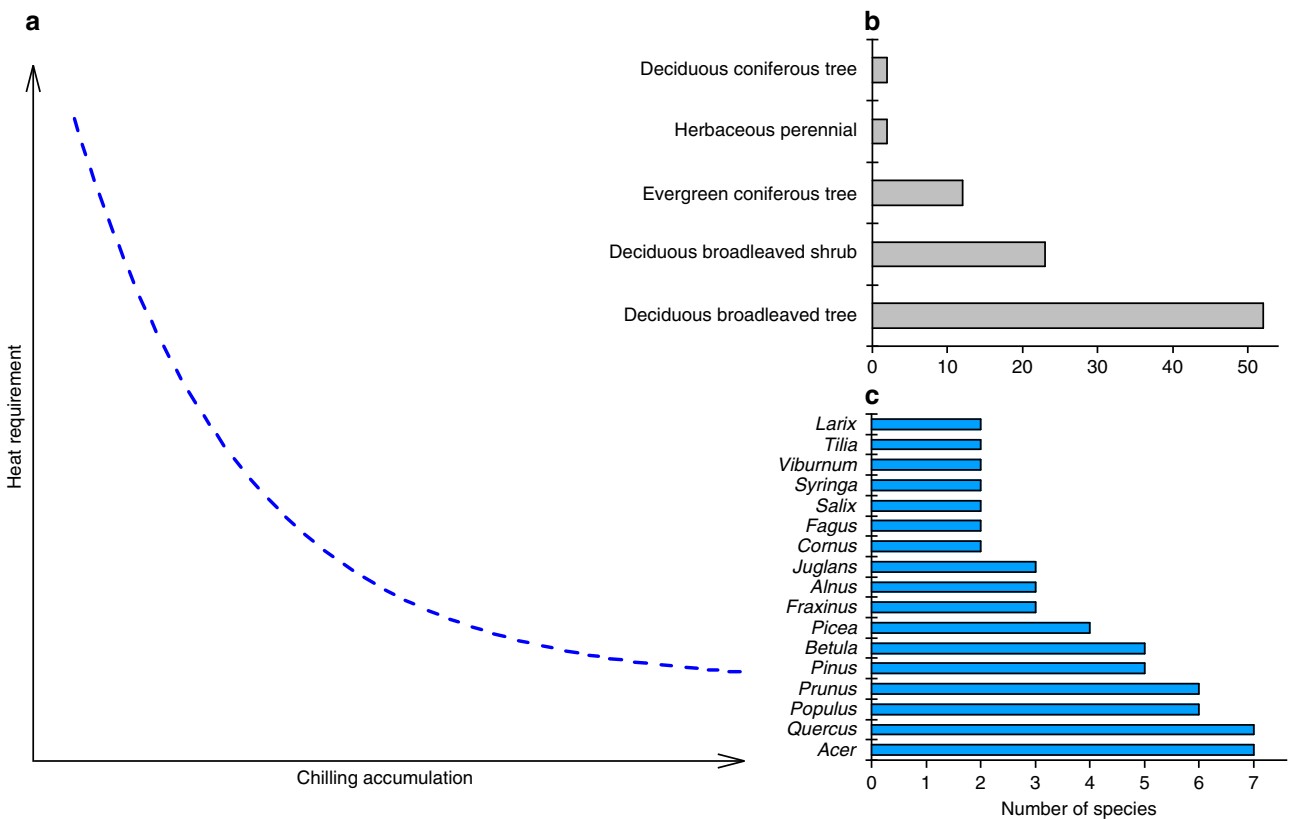

**Fig. 1 Conceptual scheme of the relationship between HR and CA.** Previous studies (Supplementary Table 1) found that saplings and twigs of 91 perennials need less heat for budburst (or leaf-out and flowering) if they receive more or longer chilling in natural or controlled environments. **a** Chilling accumulation (CA) is linearly or nonlinearly (the figure only shows a nonlinear relationship) negatively correlated with heat requirement (HR). The subplot shows the number of species exhibiting such a relationship for each life form (**b**) and genus (**c**). The CA–HR relationships for each species are shown in Supplementary Fig. 1.

and *Corylus avellana* (widely grown hazelnut plants for commercial nut production) (Supplementary Fig. 7).

In order to assess the scale effect caused by large grid size of E-OBS data set (about 10 km) and the non-uniform cover of plants and temperature in the grid cells, we compared the HR and CA based on E-OBS data set and a station-based Global Historical Climatology Network (GHCN) data set. The results showed that CA or HR calculated by these two data sets was significantly correlated, and the CA–HR relationship was stable among two data sets (see the example of *B. pendula* in Supplementary Fig. 8). Thus, the scale effect did not affect the relationship between CA and HR. We also evaluate the difference caused by the different starting dates of temperature accumulation. Compared to temperature accumulated since 1 January, the later starting date (15 January and 1 February) would result in less HR (based on Model $F_1$), especially for areas with a warm winter (see the example of *B. pendula* in Supplementary Fig. 9). Although the starting dates of heat accumulation altered the slope of HR against CA (based on Model $C_1$), the significantly negative correlation between HR and CA remained.

**Phenological trends based on chilling models.** We tested the impact of the choice of the chilling model on past and predicted future trends in spring phenology by developing a process-based model incorporating the relationship between CA and HR. First, we assessed whether different chilling models could reproduce spatial gradients of spring events across warm and cold regions in Europe. Most of the species showed a similar result, so here we only show an example of leaf out for *B. pendula* (Supplementary

Fig. 10). Across different locations, the simulated mean date correlated significantly with the observed mean dates with $R^2$ ranging from 0.65 and 0.72 ($p < 0.01$) for all the models. However, when comparing the root-mean-square error (RMSE), the valid models (RMSE = 4.2–6.9 days) were overall more accurate than the invalid model (RMSE = 6.5–10.7 days), because the invalid models produce an earlier leaf-out date than observation in cold regions but later leaf-out date in warm regions. Second, we assessed whether different chilling models could reproduce the observed temporal trends of spring events in Europe. Across all species and locations, the different models reproduced similar trends with the observed data from 1980 to 2018, but the invalid models simulated an earlier spring event compared to the observed data from the 1990s to the 2010s (Fig. 4a). Also, we compare the simulated and observed trends from 1980 to 2018 at different locations for the leaf out of *B. pendula* (Supplementary Fig. 11). For all models, the simulated and observed data exhibited consistently earlier trends at most locations. However, the valid model ($R^2 = 0.15$–0.17, RMSE = 0.18–0.20 days year$^{-1}$) performed better than the invalid models ($R^2 = 0.02$–0.09, RMSE = 0.22–0.34 days year$^{-1}$) when comparing the $R^2$ and RMSE between the simulated and observed trends. The invalid model seemed to overestimate the past earlier trends in the leaf-out date (Supplementary Fig. 11).

Based on the daily future climate data, the predicted advance in spring phenology during the 2020s to the 2090s was generally stronger for the invalid than the valid chilling models for all species in scenarios for Representative Concentration Pathways (RCPs) 4.5 and 8.5 (Supplementary Figs. 12 and 13). Averaged from all species, the date of onset of spring events gradually

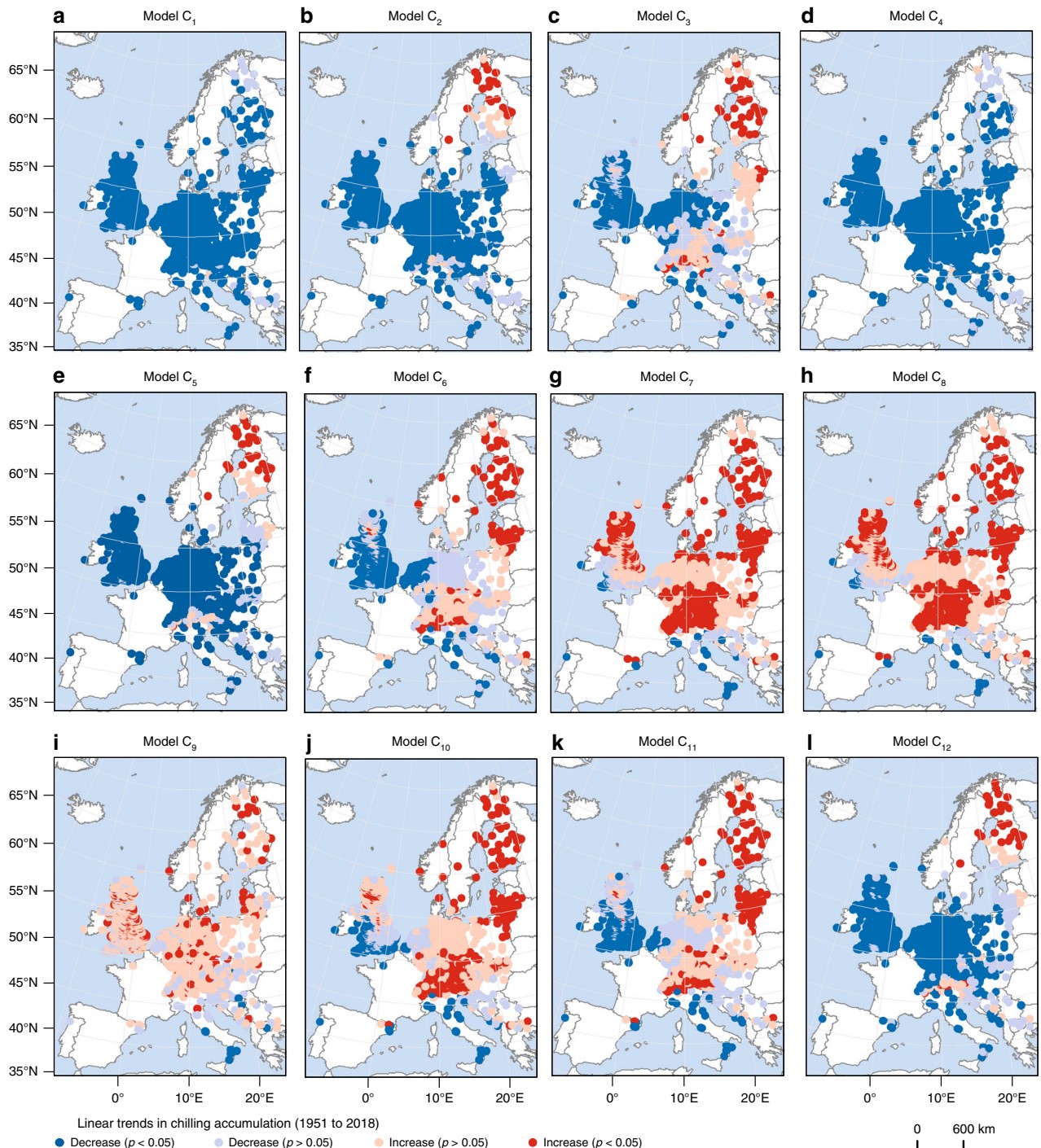

**Fig. 2 Spatial patterns of the linear trends in chilling accumulation based on 12 chilling models.** The chilling accumulation from 1 November to 30 April was calculated using the 12 chilling models. **a**–**l** Models $C_1$–$C_{12}$. Dark blue circles, significant and negative slopes (two-sided $F$-test, $p < 0.01$, $n = 68$ years). Light blue circles, insignificant and negative slopes (two-sided $F$-test, $p > 0.05$, $n = 68$ years). Dark red circles, significant and positive slopes (two-sided $F$-test, $p < 0.05$, $n = 68$ years). Light red circles, insignificant and positive slopes (two-sided $F$-test, $p > 0.05$, $n = 68$ years).

advanced until the 2060s and then remains stable under RCP 4.5 but continuously advanced under RCP 8.5 (Fig. 4). The phenological change matched the warming trends of two climate scenarios (Supplementary Fig. 14). The advance in spring events under RCP 4.5 from the 2010s to the 2090s averaged for all species was 11.2–13.7 days for the valid models but 16.8–22.5 days for the invalid models (Fig. 4c). The advances in spring events under RCP 8.5 by the end of this century averaged 20.0 days larger for the invalid than the valid models (Fig. 4d).

## Discussion

An earlier leaf-out date can increase the photosynthetic production of forests[22,23], so using an invalid chilling model to predict the start of the growing season would likely overestimate terrestrial photosynthesis and carbon uptake in spring. In addition to predicting future phenological change, the chilling models were used to estimate the chilling requirement of commercially important fruit and nut trees, such as apples, pears, and cherries[24,25]. The cultivars selected based on the amount of winter

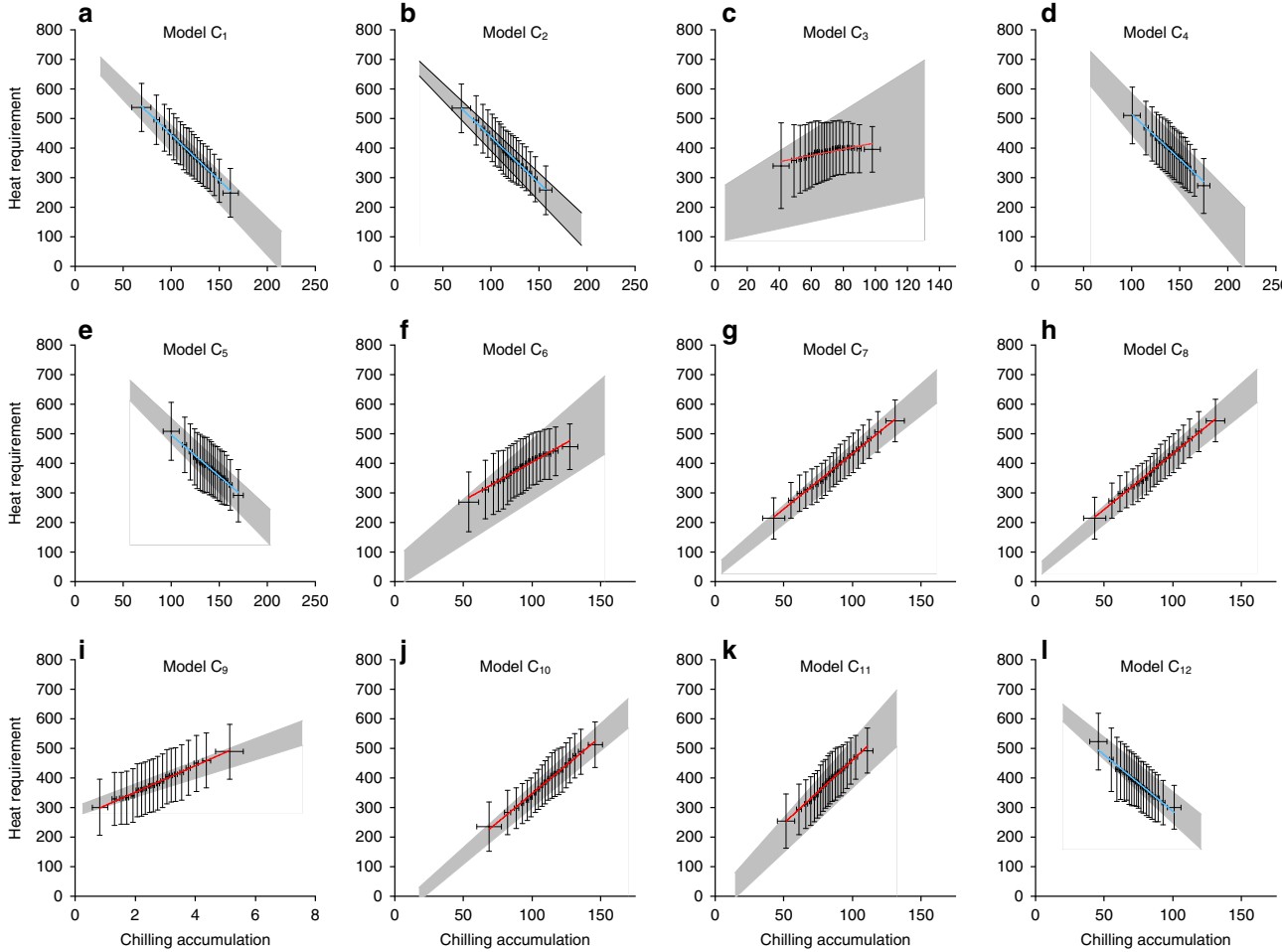

**Fig. 3 Relationship between CA and HR for leaf-out of *Betula pendula*.** Chilling accumulation (CA) from the previous 1 November to the leaf-out date was calculated using the 12 chilling models. Heat requirement (HR) was an integration of daily mean temperatures >0 °C from 1 January to the leaf-out date. **a–l** Models $C_1$–$C_{12}$. Data are divided into 20 groups with the same number of records. Error bars, mean values ± SD of HR or CA within each group ($n = 6301$ records). The linear regression lines (blue for negative correlations and red for positive correlations) are shown for each model. The shaded area shows the uncertainties, which were estimated by using the 95% confidence interval for slope and interception of regression lines.

chilling at their production sites would likely fail to grow normally in another location if an invalid model was used to estimate the adaptative capacity of CA. In this study, the amount of chilling based on invalid models even increase at mid- and high latitudes under climatic warming (Fig. 2), possibly because in these areas the daily mean temperature during the coldest period of winter was below 0 °C, which did not contribute to CA for most of the invalid chilling models (e.g., Model $C_3$ regarded that only the temperature between 0 and 5 °C was effective). Due to winter warming, the number of days with temperature below 0 °C reduced, causing an increase in the amount of chilling. If using the invalid models to estimate the amount of chilling, the increased trends in the amount of chilling would continue at mid- and high latitudes in scenarios of future climatic warming[26], so growers in these regions may miss the opportunity to adapt to climate change, e.g., by breeding tree cultivars for lower chilling requirements.

The effect of freezing temperatures is the most important structural difference between valid and invalid chilling models (Supplementary Fig. 15). The rate of chilling (the effectiveness of different temperatures on dormancy break) for the invalid models is zero or low for temperatures <0 °C. The rate of chilling for the valid models, however, is still effective even for temperatures < −5 °C. For example, if using Model $C_1$ (temperature <5 °C is

effective) to measure the amount of chilling, winter warming would lead to more number of days with the daily mean temperature higher than 5 °C, and thus decrease the CA. However, for Model $C_3$ (temperature between 0 and 5 °C is effective), this effect would be counteracted by the increase in the number of days with temperature >0 °C. Considering the increased HR of spring phenology in Europe[12], Model $C_1$ would produce a negative relationship between CA and HR, but Model $C_3$ may produce an opposite result. Thus, our study suggested that the valid models, which considered the effect of freezing temperatures on breaking dormancy, could better explain the CA–HR mechanism. Further experiments, however, are necessary to confirm this assumption, such as giving saplings or twigs a treatment of different chilling temperatures of the same duration and then a test of regrowth under the same forcing conditions[9]. However, the effectiveness of temperatures <0 °C is rarely tested before, because plants cannot survive if they are exposed to freezing temperatures immediately at the end of the growing season[27]. Tolerance to freezing under natural conditions gradually increases with exposure to low temperatures (a process known as cold acclimation), which allows plants to survive winter conditions[28]. Sudden exposure to freezing temperatures could extensively damage plant tissues due to the lack of cold acclimation (by frost cavitation). More complex experiments are

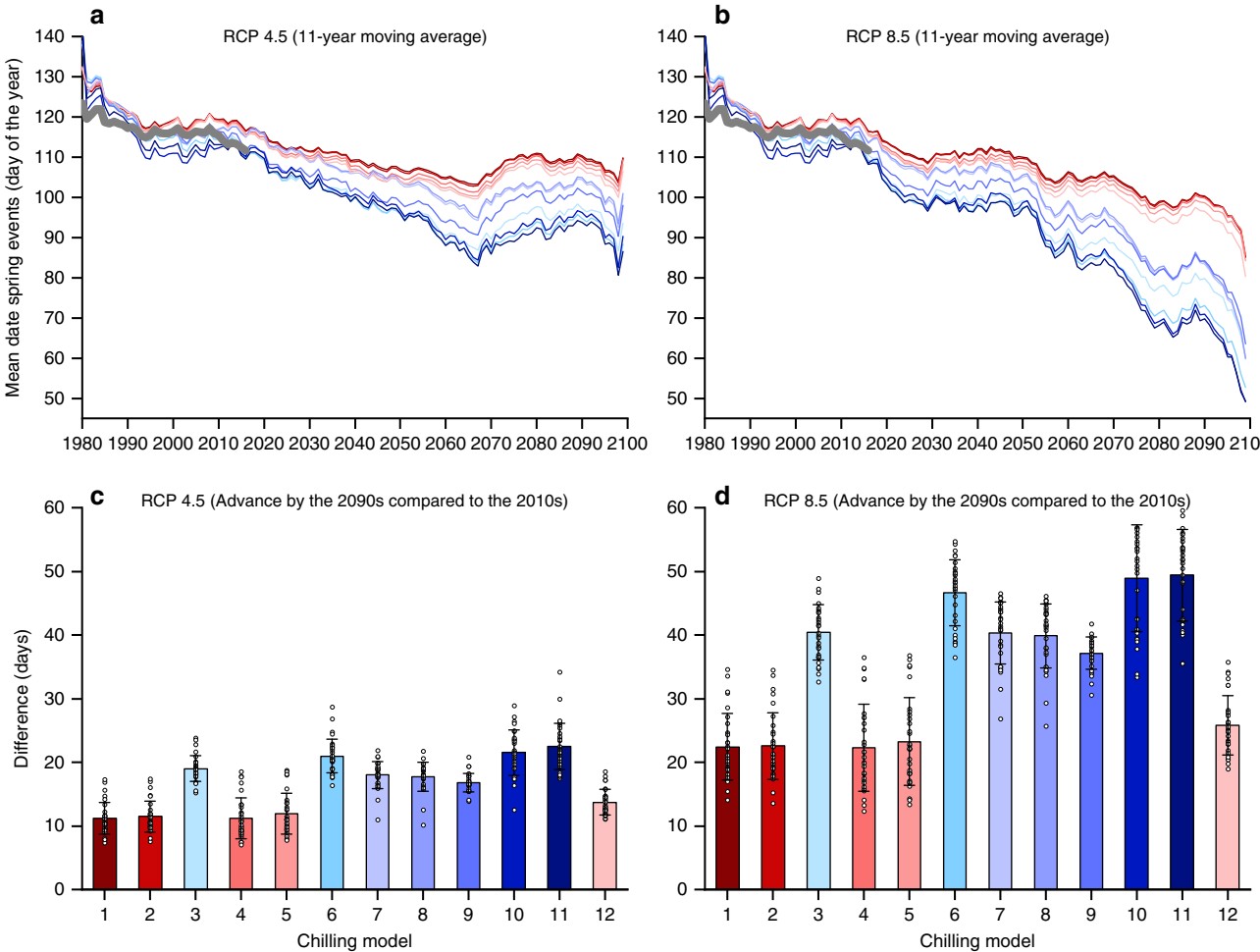

**Fig. 4 Changes in spring phenology averaged for all species and stations for 2019–2099.** Spring phenological events were simulated using linear regression between chilling accumulations based on the 12 chilling models (from 1 November to the onset of spring events) and the heat requirement (accumulated temperature >0 °C from 1 January to the onset of spring events) for all records. The observed (thick gray line) and simulated trends across spring events of all species over 1980–2018 are shown. Red colors represent valid chilling models ($C_1$, $C_2$, $C_4$, $C_5$, and $C_{12}$), and blue colors represent invalid models. **a** An 11-year moving average of spring phenological events under Representative Concentration Pathways (RCP) 4.5. **b** An 11-year moving average of spring phenological events under RCP 8.5. **c** Advance in spring events by the 2090s compared to the 2010s under RCP 4.5. **d** Advance in spring events by the 2090s compared to the 2010s under RCP 8.5. Error bars in (**c**) and (**d**), mean values ± SD ($n = 30$ species).

therefore necessary to confirm the effect of freezing temperatures on the rate of chilling, e.g., by using gradually decreasing temperatures to simulate cold acclimation in situ. Furthermore, acclimation in plants may alter the HR of plants. The acclimation could be accounted for by using different slopes and interceptions to describe the CA–HR relationship for individuals growing at different climatic conditions. However, when comparing the CA–HR relationship between low latitude (lower than 50.65° N) and high latitude (higher than 50.65° N), we did not found a significant difference in CA–HR relationship (Supplementary Fig. 16). Thus, we did not consider the impact of acclimation in this study.

Overall, the advance in budburst and leaf formation of perennials in recent decades were associated with spring warming. Due to the negative relationship between CA and HR, the reduced winter chilling slowed the spring phenological advance. We report that the majority of the current chilling models failed to represent the negative relationship between CA and HR, using 2,493,644 phenological records for 30 species from 1951 to 2018, potentially because these models were developed for limited species in specific geographic regions and did not consider the effectiveness of freezing temperature. If these invalid chilling

models are used to predict future phenological change, the advance in spring phenology at the end of this century would be twice as early as predicted using the valid models, and future spring terrestrial net carbon uptake would consequently be overestimated.

## Methods

**Phenological and climatic data**. We used data from the Pan European Phenology Project (PEP725)[29], an open-access database with long-term plant phenological observations across 25 European countries (http://www.pep725.eu/). The regional/ national network partners of PEP725 are following a consistent guideline for phenological observations[30] and prepare the data for submission to the PEP725 database curators[29]. We selected 30 species for which sufficient observational data were available: 21 deciduous broadleaved trees or shrubs, 6 herbaceous perennials, 2 evergreen coniferous trees, and 1 deciduous coniferous tree (Supplementary Table 3). Particularly, our data set included one fruit tree (*Prunus avium*) and one nut tree (*Corylus avellana*) since some of the chilling models are specifically developed for fruit and nut trees. A total of 2,493,644 individual records from 15,533 phenological stations were used. The stations were mainly distributed in moderate climates in Central Europe (Supplementary Fig. 3). Four spring events based on the BBCH code were investigated: BBCH 10, 11, 60, and 69, representing first leaves separated, first leaves unfolded, first flowers open, and end of flowering, respectively[31].

We used the E-OBS v19.0eHOM data set[32] with a spatial resolution of 0.1 × 0.1° for 1950–2018 for calculating CA and HR of the in situ phenological records. This

data set is provided by the European Climate Assessment & Data set project and includes homogenized series of daily mean, minimum, and maximum temperatures. We also use the daily maximum and minimum temperature data from the GHCN data set[33] to assess the scale effect. The GHCN data set contains station-based measurements from over 90,000 land-based stations worldwide, but only parts of PEP725 stations match with the GHCN stations.

For future climatic data (2019–2099), we used daily minimum and maximum temperatures simulated by the HADGEM2-ES model (with a spatial resolution of 0.5 × 0.5°) under two climatic scenarios (RCP 4.5 and RCP 8.5). These data have been bias-corrected by applying the method used in the Inter-Sectoral Impact Model Intercomparison Project (ISIMIP)[34], which were available on the ISIMIP server (https://esg.pik-potsdam.de/projects/isimip2b/).

**Chilling models**. We used 12 chilling models to measure the amount of chilling. One type of chilling model is based on several specific temperature thresholds. The most commonly used model, developed in the 1930s and 1940s for peach[35], calculates the number of hours or days with temperatures <7.2 or 7 °C. Another commonly used upper-temperature threshold for chilling is 5 °C[11,18]. Some studies, however, have suggested that subfreezing temperatures were effective[36,37], so we also tested chilling models using −10 °C as the lower limit. Many studies have also assumed that freezing temperatures did not contribute to winter CA and only included temperatures >0 °C for calculating CA[11,38]. Models C$_1$–C$_6$ were developed based on various combinations of the upper and lower temperature limits. The rate of chilling was 1 for daily temperatures <5 °C for Model C$_1$ (Eq. (1)), between −10 and 5 °C for Model C$_2$ (Eq. (2)), between 0 and 5 °C for Model C$_3$ (Eq. (3)), <7 °C for Model C$_4$ (Eq. (4)), between −10 and 7 °C for Model C$_5$ (Eq. (5)), and between 0 and 7 °C for Model C$_6$ (Eq. (6)). The equations for Models C$_1$–C$_6$ are as follows:

$$CU_1 = \begin{cases} 1 & T \le 5 \\ 0 & T > 5 \end{cases}, \qquad (1)$$

$$CU_2 = \begin{cases} 1 & -10 \le T \le 5 \\ 0 & T > 5 \text{ or } T < -10 \end{cases}, \qquad (2)$$

$$CU_3 = \begin{cases} 1 & 0 \le T \le 5 \\ 0 & T > 5 \text{ or } T < 0 \end{cases}, \qquad (3)$$

$$CU_4 = \begin{cases} 1 & T \le 7 \\ 0 & T > 7 \end{cases}, \qquad (4)$$

$$CU_5 = \begin{cases} 1 & -10 \le T \le 7 \\ 0 & T > 7 \text{ or } T < -10 \end{cases}, \qquad (5)$$

$$CU_6 = \begin{cases} 1 & 0 \le T \le 7 \\ 0 & T > 7 \text{ or } T < 0 \end{cases}, \qquad (6)$$

where CU$_i$ is the rate of chilling for Model C$_i$, and $T$ is the daily mean temperature (°C).

Model C$_7$ is also known as the Utah Model[39], which assigned different weights to different ranges of temperatures and was first used to measure the chilling requirements of peach (Eq. (7)). The Utah Model was modified to produce Model C$_8$ (Eq. (8)), which removed the negative contributions of warm temperatures to accumulated chilling[20].

$$CU_7 = \begin{cases} 0 & T \le 1.4 \\ 0.5 & 1.4 < T \le 2.4 \\ 1 & 2.4 < T \le 9.1 \\ 0.5 & 9.1 < T \le 12.4 \\ 0 & 12.4 < T \le 15.9 \\ -0.5 & 15.9 < T \le 18 \\ -1 & T > 18 \end{cases}, \qquad (7)$$

$$CU_8 = \begin{cases} 0 & T \le 1.4 \\ 0.5 & 1.4 < T \le 2.4 \\ 1 & 2.4 < T \le 9.1 \\ 0.5 & 9.1 < T \le 12.4 \\ 0 & T > 12.4 \end{cases}, \qquad (8)$$

where CU$_i$ is the rate of chilling for Model C$_i$, and T is the daily mean temperature (°C).

Model C$_9$ is a dynamic model developed for peach in Israel and South Africa[40,41] and now adopted for apricot cultivars[42]. The most important characteristic of Model C$_9$ was that a previous intermediate product affected the rate of chilling in the current hour or day. We did not provide equations for Model C$_9$ for simplicity (see the equation in Luedeling et al.[20]).

Harrington et al.[43] summarized published results for chilling units and constructed a chilling function based on a three-parameter Weibull distribution, coded as Model C$_{10}$ (Eq. (9)). Model C$_{11}$ has a triangular form, which was fitted by

Hänninen[44] using previous experimental results for Finnish birch seedlings (Eq. (10)). Zhang et al.[45] recently fitted observational data to the triangular model for 24 plant species and found that a mean optimal chilling temperature of 0.2 °C and an upper limit of the chilling temperature of 6.9 °C were most effective. Model C$_{12}$, therefore, uses the triangular form with parameters of 0.2 and 6.9 °C (Eq. (11)).

$$CU_{10} = \begin{cases} 1 & 2.5 < T < 7.4 \\ 0 & T < -4.7 \text{ or } T > 16 \\ 3.13\left(\frac{T+4.66}{10.93}\right)^{2.10} e^{-\left(\frac{T+4.66}{10.93}\right)^{3.10}} & \text{else} \end{cases}, \qquad (9)$$

$$CU_{11} = \begin{cases} 0 & T \le -3.4 \text{ or } T \ge 10.4 \\ \frac{T+3.4}{5+3.4} & -3.4 < T \le 5 \\ \frac{T-10.4}{5-10.4} & 5 < T < 10.4 \end{cases}, \qquad (10)$$

$$CU_{12} = \begin{cases} 0 & T \le -6.5 \text{ or } T \ge 6.9 \\ \frac{T+6.5}{6.9-0.2} & -6.5 < T \le 0.2 \\ \frac{6.9-T}{6.9-0.2} & 0.2 < T < 6.9 \end{cases}, \qquad (11)$$

where CU$_i$ is the rate of chilling for Model C$_i$, and $T$ is the daily mean temperature in °C.

**Forcing models**. Forcing models were used to measure HR for the spring events of plants. The GDD model is the most commonly used forcing model, which assumes that the rate of forcing is linearly correlated with temperature if the temperature is above a particular threshold. We mainly used Model F$_1$ (Eq. (12)), which adopts a temperature threshold of 0 °C[46–48], for examining the relationship between CA and HR:

$$FU_1 = \max(T, 0), \qquad (12)$$

where FU$_1$ is the rate of forcing for Model F$_1$, and $T$ is the daily mean temperature (°C).

We also validated the chilling models by correlating them with HR based on seven other forcing models to assess the impact of the choice of forcing model on the results. Model F$_2$ (Eq. (13)) is also a GDD model but has a temperature threshold of 5 °C[12,18]:

$$FU_2 = \max(T - 5, 0), \qquad (13)$$

where FU$_2$ is the rate of forcing for Model F$_2$, and $T$ is the daily mean temperature (°C).

Piao et al.[48] found that leaf onset in the Northern Hemisphere was triggered more by daytime than nighttime temperature. They thus proposed a GDD model using maximum instead of mean temperature. Models F$_3$ (Eq. (14)) and F$_4$ (Eq. (15)) are based on maximum temperature with thresholds of 0 and 5 °C, respectively.

$$FU_3 = \max(T_{\max}, 0), \qquad (14)$$

$$FU_4 = \max(T_{\max} - 5, 0), \qquad (15)$$

where FU$_i$ is the rate of forcing for Model F$_i$, and $T_{\max}$ is the daily maximum temperature (°C).

A recent experiment demonstrated that the impact of daytime temperature on leaf unfolding for temperate trees was approximately threefold higher than the impact of nighttime temperature[49]. Model F$_5$ (Eq. (16)) thus uses two parameters (0.75 and 0.25) to weigh the impact of daytime and nighttime temperatures on HR.

$$FU_5 = 0.75 \times \max(T_{\max} - 5, 0) + 0.25 \times \max(T_{\min} - 5, 0), \qquad (16)$$

where FU$_5$ is the rate of forcing for Model F$_5$. $T_{\max}$ and $T_{\min}$ are the daily maximum and minimum temperatures (°C), respectively.

Many studies have suggested that the rate of forcing followed a logistic function of temperature[44,50]. Model F$_6$ (Eq. (17)) uses a logistic function proposed by Hänninen[44], and Model F$_7$ (Eq. (18)) uses another logistic function proposed by Harrington et al.[43].

$$FU_6 = \begin{cases} \frac{28.4}{1+e^{-0.185(T-18.5)}} & T > 0 \\ 0 & \text{else} \end{cases}, \qquad (17)$$

$$FU_7 = \frac{1}{1 + e^{-0.47T+6.49}}, \qquad (18)$$

where FU$_i$ is the rate of forcing for Model F$_i$, and $T$ is the daily mean temperature (°C).

Model F$_8$ is a growing degree hour (GDH) model, where species have an optimum temperature for growth and where temperatures above or below that optimum have a smaller impact[51]. Model F$_8$ (Eq. (19)) was first designed for calculating HR at hourly intervals, but we applied it at a daily interval. The stress factor in the original GDH model was ignored, because we assumed that the plants

were not under other stresses.

$$\mathrm{FU}_8 = \begin{cases} 0 & T < T_L \text{ or } T > T_c \\ \frac{T_u - T_L}{2}\left(1 + \cos\left(\pi + \pi\frac{T - T_L}{T_u - T_L}\right)\right) & T_L \ge T \ge T_u \\ (T_u - T_L)\left(1 + \cos\left(\frac{\pi}{2} + \frac{\pi}{2}\frac{T - T_u}{T_c - T_u}\right)\right) & T_u < T \le T_c \end{cases}, \quad (19)$$

where $\mathrm{FU}_8$ is the rate of forcing for Model $F_8$, T is the daily mean temperature (°C), $T_L = 4$, $T_u = 25$, and $T_c = 36$.

**Analysis**. We assessed the ability of each chilling model to represent long-term trends in the chilling conditions by calculating CA using each chilling model for each station for 1951–2018. CA was calculated as the sum of $CU_i$ from 1 November in the previous year to 30 April. The trend of CA at each station was visualized as the slope of the linear regression of CA against year. We also calculated Pearson's $r$ between each pair of chilling models for each station to determine if the chilling models were interrelated.

We calculated HR and CA of spring events for each species, station, and year to determine if the chilling models are consistent with the physiological assumption that the reduction in chilling would increase HR (Fig. 1). HR was calculated as the sum of $FU_i$ from 1 January to the date of onset of spring events using Model $F_1$, and the performances of the other forcing models ($F_2$–$F_8$) were also tested. We also compared 1 January with the other two starting dates of temperature accumulation (15 January and 1 February) to test any potential difference causing by the date when temperature accumulation begins.

CA was calculated as the sum of $CU_i$ from 1 November in the previous year to the date of onset of spring events. We chose 1 November as the start date for CA because the endodormancy of temperate trees began around 1 November[52]. We only tested the linear relationship because the data were better fitted by the linear regression than the exponential model (Fig. 3), even though CA was linearly or nonlinearly negatively correlated with HR[17]. Pearson's $r$ between CA and HR for all records was calculated for each species, with a significantly negative Pearson's $r$ ($p < 0.05$) indicating that the chilling model met the physiological assumption. We also analyzed the relationship between CA and HR at stations with at least 15-year records to determine if the results were robust at the station level.

The above analysis is based on the E-OBS data set. Given the large grid size of the E-OBS data set (about 10 km) and the non-uniform cover of plants and temperature in the grid cells, we assessed the scale effect by comparing the HR and CA based on E-OBS data set with GHCN data set. We only retained the phenological station where a corresponding meteorological station (distance and altitude difference should be less than 5 km and 100 m, respectively) existed in the GHCN data set and compared the HR, CA, and their relationship.

We developed an empirical model based on the linear regression function between CA and HR (e.g., the linear fitted line in Fig. 3) to simulate the past and future spring phenological change. We calculated CA (from the previous 1 November to the current date) and heat accumulation (from 1 January to the current date) for each species in each year using a daily step. HR for the current date was calculated using the predefined linear regression function between HR and CA. The day when heat accumulation began to be larger than HR was determined as the date of onset of spring events. Compared to our empirical models, in current terrestrial biosphere models, the simulation of leaf onset is usually only based on GDDs[53] (e.g., Model $F_1$ and $F_2$ in this study), while only one model (ORCHIDEE) consider the effect of chilling[54] (Model $C_1$ in this study). Thus, at least currently, the CA–HR mechanism has not been well represented in ecosystem models. Our modeling efforts could timely provide the basis for a better representation of vegetation phenology in terrestrial biosphere models.

We predicted the annual spring phenological change (2019–2099) for each species using the above process-based models under RCP 4.5 and RCP 8.5. To produce a consistent phenological time series from past to future, we simulated the past spring phenological change from 1980 to 2018 by using the E-OBS v19.0eHOM data set, which was resampled to the same spatial resolution (0.5 × 0.5°) with the climatic data projected by HADGEM2-ES. First, we assessed whether different chilling models could reproduce spatial gradients of spring events across warm and cold regions in Europe. For each species, the simulated mean date (1980–2018) was correlated to the observed mean dates across grids with the observation data. Second, we assessed whether different chilling models could reproduce temporal trends of spring events in Europe. For each grid with at least 15-year observation data, the simulated and observed trends were estimated as the slope of the linear regression of spring phenology against year. At last, we compared the simulated trends estimated by each chilling model with the observed trends.

**Reporting summary**. Further information on research design is available in the Nature Research Reporting Summary linked to this article.

## Data availability
Phenological data can be accessed from http://www.pep725.eu/. E-OBS data set can be accessed from https://www.ecad.eu/download/ensembles/download.php. GHCH data set can be accessed from https://www.ncdc.noaa.gov/data-access/land-based-station-data/ land-based-datasets/global-historical-climatology-network-ghcn. Future climate data (2019–2099) were available on https://esg.pik-potsdam.de/projects/isimip2b/.

## Code availability
All Matlab scripts, from the initial processing of data sets to final analyses, are archived online at https://ww2.mathworks.cn/matlabcentral/fileexchange/74606-code-for-chilling-and-forcing-model.

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

## Acknowledgements

This work was funded by the National Key R&D Program of China (Grant No. 2018YFA0606102), National Natural Science Foundation of China (Grant No. 41871032), Youth Innovation Promotion Association, CAS (Grant No. 2018070), and Program for 'Kezhen' Excellent Talents in IGSNRR, CAS (Grant No. 2018RC101). PC and JP were supported by ERC Synergy grant ERC-SyG-2013-610028 IMBALANCE-P.

## Author contributions

H.W., C.W., and Q.G. designed the research. H.W. analyzed the data and wrote the first draft of the paper. C.W., J.P., and P.C. extensively revised the writing. All the authors contributed to writing the paper.

## Competing interests

The authors declare no competing interests.
