## [Peer Review File · Nature Communications]

REVIEWER COMMENTS

Reviewer #2 (Remarks to the Author):

The question of whether warmer winters will reduce chilling accumulation (CA) and therefore delay spring phenology, even as heat requirement (HR) increases, has received some attention before. But overall, this phenomenon is not often considered in models of future plant performance. As such, I was interested in reading the manuscript.

While the authors do a good job of demonstrating variation between chilling models and their predictions for spring phenology, everything in the paper is predicated on the negative relationship between CA and HR demonstrated in Figure 1. The concern though is that Figure 1 is a conceptual figure, showing one possible relationship between CA and HR. The authors state that the data supporting the generality of this Figure can be found in the SI. And while there is indeed a table in the SI listing studies (S1), nowhere does the table indicate the findings of the papers. These are instead summarized in Table S2, which lists the number of species in different groupings that show a negative CA-HR relationship. But this is far from adequate for building the argument on which the whole paper rests. What do those relationships look like? What are the slopes? And what about the 1/3 of the data that does not follow the "expected" relationship? I don't see how you can dismiss models that show a positive relationship between CA and HR when up to 1/3 of the data in the literature might show that same relationship.

In the end, the paper is therefore a lot of modeling that shows that models differ, which we know. Without firmly basing the assumption in Figure 1 on quantitative data, the rest is an interesting exercise, but not a high impact paper.

Reviewer #3 (Remarks to the Author):

The authors present a manuscript synthesizing observational records of leaf-out and flowering of perennial plants across Europe to assess the relationship between spring phenology and winter temperatures. They argue that because of an average increase in air temperatures during winter the chilling requirement for plants are not regularly being met which could counteract the direct influence of spring warming on pushing leaf-out dates earlier. They evaluate the assumptions made in a range of "chilling models" to explore how these compare with observations of this phenomenon and find most do not accurately depict the observed negative relationship between accumulated chilling time and heat requirement. While this represents an interesting synthesis that would be of interest to the community, I found the justification for the main thrust of the manuscript a bit unclear or weak. I don't doubt there are key takeaways from this work that are highly relevant to observational and process modeling efforts focused on understanding the role of climate on plant phenology and feedbacks, but some work is needed to strengthen the thrust of the manuscript and to help the reader better understand past examples and the authors contributions. For example, the authors cite several papers and describe the main relationship between chilling and heat requirement but never provided a clear example of this, showed the empirical evidence, or clearly justified the overall theory. It would have been helpful to provide this in addition to or instead of Figure 1, which I did not find all that supportive of the analysis. I would recommend editing the argument and justification to provide more strength to the analysis and more clearly illustrate the phenological paradigm they are presenting.

Abstract:

The first three sentences require a large leap that the statements are accurate without strong justification for them. The authors then report that most models for dormancy chilling are "invalid" but don't adequately explain what exactly is meant by this statement, after which the rest of the abstract

hinges on this core statement.

Introduction

The authors should be more specific about what is meant by "mid- to high-latitudes"

The authors purport in their manuscript that winter warming has reduced chilling requirements, impacting spring phenology. However, it is not clear by their argument or the evidence that this is a universal trend in the mid- to high-latitudes? Does this generally reflect all longer-lived plants or are there cases where this paradigm does not fit the data? The authors state on line 59-62 that "at least 95 perennials..." follow the negative relationship between chilling accumulation and heat requirement (CA-HR) and provide a table with species. How much of the European land mass do these species represent? They appear to be many of the key tree species.

Not providing the various examples of the CA-HR relationship or the strengths of those relationships makes it rather difficult to assess how strong we would expect the modeled relationships to be. Does this vary by study, species, location? Can the authors provide examples of this relationship? Can this be provided independent from the models they are evaluating?

What about acclimation in plants? How is this accounted for or could be accounted for in the CA-HR paradigm?

The focus of the paper appears to hinge on a theory (CA-HR) proposed by a paper focused on a single evergreen needle-leaf tree species. Is this response general across perennials/plants?

Results

Why does Figure 1 only show the "non-linear" relationship when the authors argue there are both linear and non-linear responses, and the model examples all show the linear response?

How does the observable impact the study and how is this accounted for in the modeling? That is, differences in defining spring phenology and the observed change, from flowering, to bud-burst, to leaf unfolding.

How are uncertainties accounted for in the analysis? Are the data standardized by observer?

How are the differences in when temperature accumulation begin accounted for in the analysis? Is this part of the evaluation of mis-match with the observations? Did the authors test both the basic assumptions of the models but also try to standardize between and test their functional form as well? Or was the evaluation strictly focused on comparing how the models were implemented in their respective studies? How many of these models are currently used in ecosystem or Earth system models?

How are scale effects in the comparisons accounted for? Given the large grid size and non-uniform cover of plants and temperature in the grid cells, this could impact the relationships between models and data.

Is it that surprising that the two most extreme RCPs also result in the largest change in spring phenology?

Discussion

I don't follow the statements on lines 233-236. The authors state the observing the CA-HR relationship from reporting stations is dependent on the model used, but I had thought the authors were providing observations of this relationship to test models? How then are the correct relationships assessed if they depend on invalid models?

Point to point response to reviewers

Reviewer #2

The question of whether warmer winters will reduce chilling accumulation (CA) and therefore delay spring phenology, even as heat requirement (HR) increases, has received some attention before. But overall, this phenomenon is not often considered in models of future plant performance. As such, I was interested in reading the manuscript.

While the authors do a good job of demonstrating variation between chilling models and their predictions for spring phenology, everything in the paper is predicated on the negative relationship between CA and HR demonstrated in Figure 1. The concern though is that Figure 1 is a conceptual figure, showing one possible relationship between CA and HR. The authors state that the data supporting the generality of this Figure can be found in the SI. And while there is indeed a table in the SI listing studies (S1), nowhere does the table indicate the findings of the papers. These are instead summarized in Table S2, which lists the number of species in different groupings that show a negative CA-HR relationship. But this is far from adequate for building the argument on which the whole paper rests. What do those relationships look like? What are the slopes? And what about the 1/3 of the data that does not follow the "expected" relationship? I don't see how you can dismiss models that show a positive relationship between CA and HR when up to 1/3 of the data in the literature might show that same relationship.

In the end, the paper is therefore a lot of modeling that shows that models differ, which we

know. Without firmly basing the assumption in Figure 1 on quantitative data, the rest is an interesting exercise, but not a high impact paper.

Response: Thank you for your very important and constructive comment. We agree with this suggestion, so we have now revised our study accordingly. In the revised manuscript, we added **Supplementary Fig. 1** to provide empirical evidence of the negative relationship between chilling accumulation (CA) and heat requirement (HR) for ALL 95 species from previous studies. Except for 4 species, 91 (95.8%) perennials belonging to 46 genera followed the negative relationship. For some species, the original references did not report the curve but draw the conclusion that larger CA would decrease HR) with different slopes (linear or nonlinear). We therefore used **"Negative relationship confirmed"** (NRC) in those subfigures.

From **Supplementary Fig. 1**, we can see that the negative correlation between CA and HR showed to be highly consistent among perennials, and this relationship is stable across studies, species, and locations. In the previous version, what we presented were that 2/3 of 30 species investigated in our study had been studied before (to show the coverage of species), rather than that 1/3 of the data in the literature showed a different CA-HR (i.e., positive) relationship. In order to avoid misunderstanding by the readers, we have removed this expression in the revised manuscript.

Reviewer #3

The authors present a manuscript synthesizing observational records of leaf-out and flowering of perennial plants across Europe to assess the relationship between spring phenology and winter temperatures. They argue that because of an average increase in air temperatures during winter the chilling requirement for plants are not regularly being met which could counteract the direct influence of spring warming on pushing leaf-out dates earlier. They evaluate the assumptions made in a range of “chilling models” to explore how these compare with observations of this phenomenon and find most do not accurately depict the observed negative relationship between accumulated chilling time and heat requirement. While this represents an interesting synthesis that would be of interest to the community, I found the justification for the main thrust of the manuscript a bit unclear or weak. I don't doubt there are key takeaways from this work that are highly relevant to observational and process modeling efforts focused on understanding the role of climate on plant phenology and feedbacks, but some work is needed to strengthen the thrust of the manuscript and to help the reader better understand past examples and the authors contributions. For example, the authors cite several papers and describe the main relationship between chilling and heat requirement but never provided a clear example of this, showed the empirical evidence, or clearly justified the overall theory. It would have been helpful to provide this in addition to or instead of Figure 1, which I did not find all that supportive of the analysis. I would recommend editing the argument and justification to provide more strength to the analysis and more clearly illustrate the phenological paradigm they are presenting.

Response: Thank you for these positive comments. We have now revised the manuscript accordingly. We have added Supplementary Fig. 1 to provide empirical evidence of the negative relationship between chilling accumulation (CA) and heat requirement (HR) for 95 species from previous studies. Except for 4 species, 91 (95.8%) perennials belonging to 46 genera followed the negative relationship with different slopes (linear or nonlinear), although for some species, the original references did not report the curve but only draw the conclusion that larger CA would decrease HR. The detailed subfigures from previous references in the new supplementary Figure 1 provide more strength to the analysis.

Abstract:

The first three sentences require a large leap that the statements are accurate without strong justification for them. The authors then report that most models for dormancy chilling are “invalid” but don’t adequately explain what exactly is meant by this statement, after which the rest of the abstract hinges on this core statement.

Response: Yes, we did not give a clear meaning of ‘invalid’ here. We have now revised the Abstract following to solve this insightful concern. We have now clearly stated that the expected paradigm was a negative CA-HR relationship. More than half (7 of 12) of current chilling models are invalid since they showed a positive relationship between CA and HR (Line 26-30).

Introduction

The authors should be more specific about what is meant by “mid- to high-latitudes”

Response: We have now specified: areas north of 30°N (Line 38).

The authors purport in their manuscript that winter warming has reduced chilling requirements, impacting spring phenology. However, it is not clear by their argument or the evidence that this is a universal trend in the mid- to high-latitudes? Does this generally reflect all longer-lived plants or are there cases where this paradigm does not fit the data?

The authors state on line 59-62 that “at least 95 perennials...” follow the negative relationship between chilling accumulation and heat requirement (CA-HR) and provide a table with species. How much of the European land mass do these species represent?

They appear to be many of the key tree species. Not providing the various examples of the CA-HR relationship or the strengths of those relationships makes it rather difficult to assess how strong we would expect the modeled relationships to be. Does this vary by study, species, location? Can the authors provide examples of this relationship? Can this be provided independent from the models they are evaluating?

Response: As mentioned above, we have now added Supplementary Fig. 1 to provide empirical evidence of the negative relationship between chilling accumulation (CA) and heat requirement (HR) for 95 species from previous studies. Except for 4 species, 91 (95.8%) perennials belonging to 46 genera followed the negative relationship. Thus, more

than 95% of studied species support the negative relationship between chilling accumulation and heat requirement (CA-HR), and these species involve the most dominant forest tree species in Europe (Supplementary Fig. 2).

The negative correlation between CA and HR is thus common among perennials and stable across studies, species, and locations. This is a universal trend in the mid- to high-latitudes since this relationship has been found in the USA, Canada, European countries, and China (Supplementary Table 1). This relationship is partially independent of the models we are evaluating since previous studies used many chilling and forcing models (some are different from the models we are evaluating) to describe the CA-HR relationship. In Supplementary Fig. 1, the methods to measure CA include the number of chilling days in a specific temperature range during different periods, number of chilling hours, model C_{11} in this study. The methods to measure HR include growing degree days with a specific threshold temperature and various starting dates, growing degree hours, and days to budburst. These diverse approaches strengthen our confidence in the negative CA-HR relationship.

What about acclimation in plants? How is this accounted for or could be accounted for in the CA-HR paradigm?

Response: In this study, we did not consider the impact of acclimation. We agree with you that acclimation may alter the HR of plants. We suggest that the acclimation could be accounted for by using different slopes and interceptions to describe the CA-HR

relationship for individuals growing at different climatic stations. We compared the CA-HR relationship between low latitude (lower than 50.65 °N) and higher latitude (higher than 50.65 °N) and did not find a significant difference in CA-HR relationship (Supplementary Fig. 16). Thus, we did not consider the impact of acclimation in this study (Line 247-253).

Fig. 16 Relationship between chilling accumulation (CA) and heat requirement (HR) for leaf-out of *Betula pendula* at different latitudes. a, at latitudes lower than 50.65 ° N. b, at latitude higher than 50.65 ° C. The red line represents the linear regression line. *: $p < 0.01$.

The focus of the paper appears to hinge on a theory (CA-HR) proposed by a paper focused on a single evergreen needle-leaf tree species. Is this response general across perennials/plants?

Response: The theory (CA-HR) has been proposed by surveying previous studies since the 1980s (Supplementary Table 1). We were unclear in the earlier version. In the revised paper, we cite Supplementary Table 1 rather than just one paper. As mentioned above and in supplementary Figure 1, this response is general across perennials/plants.

Results

Why does Figure 1 only show the “non-linear” relationship when the authors argue there are both linear and non-linear responses, and the model examples all show the linear response?

Response: Thank you for this comment. As shown in Supplementary Fig. 1, there are both linear and non-linear responses for previous studies, which may be partly caused by models, study time durations, and experimental designs. However, in our *in-situ* data, for all species, we found that data were better fitted by the linear regression than the exponential model (data shown in Figure 3). Therefore, we used the linear response for the comparison among models.

How does the observable impact the study and how is this accounted for in the modeling?

That is, differences in defining spring phenology and the observed change, from flowering, to bud-burst, to leaf unfolding. How are uncertainties accounted for in the analysis? Are the data standardized by observer?

Response: Thank you for this important suggestion, and here are our clarifications that may help to alleviate concerns from observation.

First, in the PEP725 data, the definition of spring phenology was constant. All phenological observation networks in Europe are following a guideline for phenological observations (Koch et al. 2009). The regional/national network partners then prepare the

data for submission to the PEP725 database curators. Dates are converted to day-of-the-year values and phenophase codes translated to the BBCH scale (Meier 2001). Please find a detailed introduction about the phenological data in Templ et al (2018). We have now added more detailed information about the dataset in the revised manuscript (Line 268-269).

Second, we agree with you that within a year, the plant developmental stage will change from flowering to bud-burst, to leaf unfolding (or the other order). However, the phenological observers record these events separately with the respective definition. For example, BBCH 10, first leaves of coniferous tree separated; BBCH 11, first leaves of broadleaved trees unfolded; BBCH 60, first flowers open; and BBCH 69, end of flowering (Meier 2001). Thus, the observed change in phenology did not affect our results.

Third, we used the original data and the observer did not standardize the data. In phenological studies, phenological data represent a specific date and is usually not standardized. For example, Menzel et al. 2006 concluded a clearly visible shift of most spring phases to earlier entry dates since 1951 in Europe based on the same database used in our study (not standardized). The most important reason is that standardization will not affect the interannual variability for the CA-HR relationship, and therefore, these data have been extensively used recently in such long temporal analyses. Other examples using the raw PEP725 data include Guan 2014; Piao et al 2015, Fu et al. 2015, and many others.

Finally, the uncertainties of the CA-HR relationship were estimated by using the 95% confidence interval for slope and interception of the linear regression model. Please see the shaded area of the linear regression line in Fig.3.

References:

Koch E, Bruns E, Chmielewski F, et al. Guidelines for plant phenological observations.

Geneva: World Meteorological Organization, 2009.

Meier U. Growth stages of mono-and dicotyledonous plants: BBCH Monograph, 2nd ed.

Berlin: Federal Biological Research Centre for Agriculture and Forestry, 2001.

Templ B, Koch E, Bolmgren K, et al. Pan European Phenological database (PEP725): a

single point of access for European data. *International Journal of Biometeorology*, 2018, 62(6): 1109-1113.

Menzel A, Sparks T H, Estrella N, et al. European phenological response to climate

change matches the warming pattern. *Global Change Biology*, 2006, 12(10): 1969-1976.

Guan B T. Ensemble empirical mode decomposition for analyzing phenological responses

to warming. *Agricultural and Forest Meteorology*, 2014, 194: 1-7.

Piao S, Tan J, Chen A, et al. Leaf onset in the northern hemisphere triggered by daytime

temperature. *Nature Communications*, 2015, 6: 6911.

Fu Y H, Zhao H, Piao S, et al. Declining global warming effects on the phenology of spring

leaf unfolding. *Nature*, 2015, 526(7571): 104-107.

How are the differences in when temperature accumulation begin accounted for in the analysis? Is this part of the evaluation of mis-match with the observations? Did the authors test both the basic assumptions of the models but also try to standardize between and test their functional form as well? Or was the evaluation strictly focused on comparing how the models were implemented in their respective studies? How many of these models are currently used in ecosystem or Earth system models?

Response: Thank you for these important questions. Here are our detailed explanations.

(1) In order to test any potential difference caused by the date when temperature accumulation begins, we compared 1 January with the other two starting dates of temperature accumulation, all of which were commonly used by the scientific community. As a result, Supplementary Fig. 9 was added in the revised manuscript. Compared to temperature accumulated since 1 January, the later starting date (15 January and 1 February) would result in less heat requirement (HR) (see the example of *Betula pendula* in Supplementary Fig. 9), especially for areas with a warmer winter (mean temperature $>0^{\circ}\text{C}$ in January). However, the correlation between HRs based on different starting date was significant (Supplementary Fig. 9a, b). Although the starting date of heat accumulation altered the slope of HR against CA (Supplementary Fig. 9c), the significantly negative correlation between HR and CA still remained (Line 158-163).

Supplementary Fig. 9 | Comparison between heat requirement (HR) for leaf-out of *Betula pendula* based on different starting dates of temperature accumulation. HR was calculated based on a commonly used forcing model (integrating daily mean temperatures >0 °C). **a**, HR accumulated from 1 January vs. from 15 January; **b**, HR accumulated from 1 January vs. from 1 February; **c**, chilling accumulation(CA)-HR relationship for different starting dates of temperature accumulation. CA was calculated based on model C_1 accumulated from 1 November in the previous year to the leaf-out date. The linear regression lines are shown. *: $P < 0.01$.

(2) We did not try to standardize between and test the functional form of models. We used the basic assumptions of the models and their original functional forms and then focused on comparing the performances of each model on simulating the CA-HR relationship based on our observation data. As explained in the last comment, this way is acceptable,

especially when considering that there might not to have a universal form for all these models.

(3) A previous study (Richardson et al, 2012) summarized phenological models currently used in ecosystem models or terrestrial biosphere models (see the table below). The results showed that many terrestrial biosphere models only consider the growing degree days (e.g., model F1 and F2 in this study) to predict leaf onset date and only one model (ORCHIDEE) considers the effect of chilling (model C₁ in this study). This shows that at least currently, the CR-HR has not been well represented in ecosystem models, so our modeling efforts could timely provide the basis for a better representation of vegetation phenology in terrestrial biosphere models. We have now added this information into the Methods section (Line 413-418).

The table in [Richardson A D, Anderson R S, Arain M A, et al. Terrestrial biosphere models need better representation of vegetation phenology: results from the North American Carbon Program Site Synthesis. *Global Change Biology*, 2012, 18(2): 566-584.] summarized terrestrial biosphere models and their representation of phenology and seasonality of leaf area index (LAI).

Table 2 Summary of models used in this analysis and their representation of phenology and seasonality of leaf area index (LAI) For models with 'prognostic' phenology, the seasonality of LAI is predicted based on climatic drivers; for those with 'prescribed' phenology, an average seasonal LAI cycle, as derived on a site-by-site basis from satellite (AVHRR) data, was used. Models with semi-prescribed and semi-prognostic phenology represent a hybrid of these approaches. GDD is growing degree days; T is temperature; C is carbon; PFT is plant functional type

Model name	Resolution	Leaf onset	Control on LAI	Leaf loss	Source
BEPS	Daily	Satellite	Satellite	Satellite	Ju et al. (2006)
Biome-BGC	Daily	GDD and radiation sum	Dynamic C allocation	Daylength and low temperature	Thornton et al. (2002)
Can-IBIS	Half-hourly	T threshold	GDD and dynamic C	Prescribed	El Maayar et al. (2002)
CN-CLASS	Half-hourly	C balance	C balance	Daylength and low temperature	Arain et al. (2006)
DLEM	Daily	$T_{7\text{-day}} > \text{threshold}$	GDD to PFT limit	Daylength and low temperature	Tian et al. (2010)
Ecosys	Hourly	Hours above T threshold	Dynamic C allocation	Hours below T threshold	Grant et al. (2009)
ED2	Half-hourly	Semi-prescribed	Dynamic C allocation	GDD and leaf turnover	Medvigy et al. (2009)
ISAM	Half-hourly	Prescribed	Prescribed	Prescribed	Jain & Yang (2005)
LoTEC	Half-hourly	GDD	GDD	T-dependent turnover	Hanson et al. (2004)
LPJ_wsl	Daily	GDD	GDD	Leaf longevity (prescribed)	Sitch et al. (2003)
ORCHIDEE	Half-hourly	GDD and chilling	Dynamic C allocation	Decreasing T and T threshold	Krinner et al. (2005)
SiB3	Half-hourly	Prescribed	Prescribed	Prescribed	Baker et al. (2008)
SiBCASA	10 min	Prescribed	Prescribed	Prescribed	Schaefer et al. (2008)
SSiB2	Half-hourly	Prescribed	Prescribed	Prescribed	Zhan et al. (2003)

How are scale effects in the comparisons accounted for? Given the large grid size and non-uniform cover of plants and temperature in the grid cells, this could impact the relationships between models and data.

Response: In order to assess the scale effect caused by the large grid size of E-OBS data set (about 10km) and the non-uniform cover of plants and temperature in the grid cells, we have added a new climate data into the analysis. We have compared the HR and CA based on the E-OBS data set and a station-based Global Historical Climatology Network (GHCN) data set. We only retained the phenological stations where a corresponding meteorological station (distance and altitude difference less than 5 km and 100 m) existed in the GHCN data set and compared the HR, CA, and their relationship.

We give an example of *Betula pendula* in **Supplementary Fig. 8**. The results showed that CA or HR calculated by these two datasets were significantly correlated with $R^2 > 0.9$

($P < 0.01$). The CA-HR relationship is stable among two dataset: $y = -2.81x + 718.0$ (E-OBS) vs. $y = -2.98x + 768.4$ (GHCN). Thus, the scale effect did not affect the relationship between CA and HR (Line 152-157).

Supplementary Fig. 8 | Comparison between chilling accumulation (CA) and heat requirement (HR) for leaf-out of *Betula pendula* based on two temperature dataset. CA was calculated based on model C_1 accumulated from 1 November in the previous year to the leaf-out date. HR was calculated based on a commonly used forcing model (integrating daily mean temperatures $> 0^{\circ}\text{C}$ from 1 January to leaf-out date). The CA and HR in corresponding grid cells of E-OBS data and at corresponding stations of GHCN data are compared. a, CA between two datasets; b, HR between two datasets; c, CA-HR data are compared. a, CA between two datasets; b, HR between two datasets; c, CA-HR relationship for E-OBS data; d, CA-HR relationship for GHCN data. Root-mean-square

error (RMSE) between the two datasets is shown. Also, the linear regression line and its R^2 are shown (in red). *: $P < 0.01$.

Is it that surprising that the two most extreme RCPs also result in the largest change in spring phenology?

Response: Emissions in RCP 4.5 peak around 2040, then decline, while in RCP 8.5 emissions continue to rise throughout the 21st century. As a result, the predicted winter and spring mean temperature continues to increase from 2020 to 2060, and then remain stable under RCP 4.5 but continuously increase in RCP 8.5 (Supplementary Fig. 14). The phenological change matched the warming trends of two climate scenarios. For example, the spring phenological advance stopped from 2060 in RCP 4.5 (Fig. 4a). Given that warming controlled the spring phenology, both RCPs will cause large changes in spring phenology before 2060, and even larger at RCP 8.5 (Fig 4 c and d). This is thus reasonable and worth characterizing to have the wide range of scenarios considered.

Supplementary Fig. 14 | Temperature change in Europe from 2012 to 2099. For the past period, temperature data is from the E-OBS data. For future climatic data (2019-2099), we used the data simulated by the HADGEM2-ES model. The temperature during the chilling period is averaged from the previous 1 November to 31 January, while the temperature during the forcing period is averaged from 1 February to 31 May. The bold lines show the result of the 11-year moving average.

Discussion

I don't follow the statements on lines 233-236. The authors state the observing the CA-HR relationship from reporting stations is dependent on the model used, but I had thought the authors were providing observations of this relationship to test models? How then are the correct relationships assessed if they depend on invalid models?

Response: We expressed inaccurately here in the original version. We have now clarified that as Supplementary Fig.1 shows, the negative correlation between CA and HR is common among perennials and stable across studies, species, and locations. The results of these studies are partly independent of the models we are evaluating since they use various methods (some methods are different from our models) to assess the CA-HR relationship in Europe, North America, and Asia and draw a consistent conclusion. Thus, we remove the inaccurate expression in the revised manuscript.

REVIEWERS' COMMENTS:

Reviewer #3 (Remarks to the Author):

I have reviewed the revised manuscript and have found all of my previous concerns have been thoroughly addressed. I very much appreciate all of the work that the authors put into this revision. The paper is much clearer and as a result provides much stronger support for the authors main argument, that there is an urgent need to address the representation of phenological modeling, in this case specifically chilling models.

Point-by-point response to issues raised by referees

Reviewer #3 (Remarks to the Author):

I have reviewed the revised manuscript and have found all of my previous concerns have been thoroughly addressed. I very much appreciate all of the work that the authors put into this revision. The paper is much clearer and as a result provides much stronger support for the authors main argument, that there is an urgent need to address the representation of phenological modeling, in this case specifically chilling models.

Response: Because the reviewer has no further comments, we only respond to the editorial requests.